# OpenReview forum: "Reward-guided Meta-Prompt Evolving with Reflection for LLM Jailbreaking"
_ICLR.cc/2026/Conference — ICLR 2026 Conference Withdrawn Submission_

### Official Review · Reviewer_71pz · 2025-10-27

**Soundness:** 3
**Presentation:** 3
**Contribution:** 3
**Rating:** 6
**Confidence:** 4

**Summary:**

This paper introduces ROOT (“Reward-guided meta-prompt evolving with reflection”), a fully-automated, closed-loop framework for generating jailbreak prompts against LLMs. The pipeline: a Meta LLM emits an attack prompt from a meta-prompt; a Victim LLM is queried; a Judge LLM issues a binary success label and a free-text “reflection”; an Analysis LLM summarizes strategies from reflections of successes and failures; an Optimize LLM updates the meta-prompt; iterate.

Experiments show high ASR on AdvBench and HarmBench across open- and closed-source models, plus ablations on optimization steps, strategy count, and success-vs-failure reflections. The paper claims >90% success on all tested models and near-universal bypass of guard models (e.g., OpenAI Moderation). Key artifacts include: the initialization meta-prompt (Fig. 3, p.4), the closed-loop diagram (Fig. 2, p.3), Algorithm 1 (p.6), tables of ASR results (Tables 1–3, pp.7–8), category breakdowns (Fig. 4, p.8), ablations (Figs. 5–7, pp.9,14), transferability (Table 5, p.16), and prompts for judges/optimizers (Appx. D, pp.18–20).

**Strengths:**

### 1. Clear and novel idea
The method ties generation, judging, reflection aggregation, and meta-prompt optimization into one iterative loop (Fig. 2, p.3; Algorithm 1, p.6), making the contribution easy to follow.

### 2. Strong experiment results
Achieves a mean ASR > 90% on both the AdvBench and HarmBench benchmarks, demonstrating consistent performance across different architectures such as GPT-4(o), Claude 3.5 Sonnet, and LLaMA 3 70B.

In particular, the results presented in Tables 1–3 show an average improvement of +30 to +50 points over prior automated attack methods (e.g., AutoDAN, TAP), proving that strategic meta-prompt optimization is substantially more effective

### 3. Cross-model transferability
In the cross-model transfer experiments shown in Table 5 (p. 16), ROOT maintains a high success rate even on models that it was not trained on.

This indicates that ROOT’s optimization process does not overfit to the policy of any specific LLM, but rather learns generalizable jailbreak strategies that transfer effectively across different model architectures.


### 4. Empirical Effect of Reflection-based Strategy Evolution
Figures 5 and 6 demonstrate that incorporating reflection analysis of both successful and failed cases leads to an ASR increase of +20 to +30 points.

Rather than merely counting failures, ROOT textually summarizes the reasons for failure and feeds this information back into the learning loop. This reflective feedback mechanism serves as the key contributor to performance improvement, enabling the model to evolve more effective attack strategies over time.

As shown in Figure 7 (p. 14), performance saturates after approximately three iterations of meta-prompt updates, indicating that ROOT exhibits convergent and stable optimization behavior.
Additionally, the strategy-count experiment in Figure 5 reveals that using around eight strategies provides an optimal balance between performance and computational cost, offering practical insights for system design and efficiency trade-offs.

**Weaknesses:**

### 1. Weak Baselines

Although ROOT reports high success rates — 99% on GPT models and over 90% across all models (Tables 1–3) — the defense baselines evaluated are relatively weak, relying mainly on simple moderation systems such as OpenAI Moderation and LLaMA Guard.
More robust adaptive defenses or multi-layer moderation frameworks have not been tested, raising questions about whether ROOT’s success would persist against stronger or adversarially trained defenses.

### 2. Need for Diverse Judge Combinations
The evaluation primarily depends on a single GPT-4-based judge, which may bias the reward estimation process.
It would be valuable to perform ensemble judging — combining multiple heterogeneous models (e.g., Claude, Gemini, or LLaMA-based judges) — to assess how sensitive ROOT’s optimization loop is to variations in judgment distributions and to ensure robustness against judge-specific bias.

### 3. Multi-turn
The paper does not address how a victim LLM might change its refusal or avoidance strategies (or conversely how vulnerabilities might amplify) by leveraging previous prompts and responses in repeated conversations. In other words, the cumulative effect of multi-turn context is not evaluated.

### 4. No measurement of sensitivity to role/persona changes
There is a lack of robustness analysis for multi-turn patterns in which the attacker switches roles (e.g., physician ↔ engineer) or sets different personas for the victim. The paper does not measure how sensitive ROOT’s success is to such role or persona manipulations.

I believe this paper has only minor weaknesses, and that it presents sufficiently strong experiments, results, and ideas.

**Questions:**

typo: Table 1  Victim LLLMs to Victim LLMs.

---

### Official Review · Reviewer_qQVP · 2025-10-28

**Soundness:** 2
**Presentation:** 3
**Contribution:** 2
**Rating:** 2
**Confidence:** 3

**Summary:**

This paper proposes an automatic black-box jailbreaking method named **ROOT**. The method begins with a meta-prompt containing toxic or harmful questions, which is fed into LLMs to generate jailbreak prompts. These generated prompts are then judged and evaluated by a *Judge LLM*. The feedback from the Judge LLM subsequently guides the optimization process of the meta-prompt, increasing the likelihood of successful jailbreak attacks. Experimental results show that ROOT achieves the highest attack success rates compared to other black-box jailbreaking baselines across multiple evaluation datasets.

**Strengths:**

- The paper explores a timely and relevant topic.
- The proposed methodology is highly automated and can iteratively improve itself.
- The evaluation results demonstrate that the proposed method achieves state-of-the-art performance.

**Weaknesses:**

- The proposed methodology does not fundamentally distinguish itself from existing black-box jailbreaking techniques, and thus the technical novelty is limited.
- The method heavily relies on the specific victim LLM and the provided harmful instruction, which may lead to overfitting.
- The definitions and calculations of *attack success rate* (ASR) and *bypass rate* are obscure and insufficiently explained.

**Questions:**

Thank you for submitting this work. I find the paper generally easy to follow and well-written. However, I have several concerns regarding the technical contribution, performance evaluation, and depth of analysis. I hope the following comments help improve the quality of the paper.

- The proposed methodology largely follows the design pattern of existing black-box jailbreaking methods. The authors start with a harmful prompt, then use a prompt generator and an AI judge to iteratively refine the jailbreak prompt so that the victim LLM is more likely to be compromised. This idea is not novel and has already been widely adopted by previous black-box jailbreak approaches, with only minor technical variations. In the Introduction, the authors highlight a limitation of existing methods, “the complex and discrete strategy design.” However, this issue does not seem to be effectively addressed by ROOT.
- ROOT appears to be optimized for a specific victim LLM and specific harmful instructions, rather than providing a generalizable or model-agnostic jailbreak mechanism. This dependency limits the method’s applicability.
- The evaluation section mainly focuses on *attack success rate* (ASR) and *bypass rate*. The concern here is twofold:

    (1) How do the authors define ASR? Is this metric calculated by the same Judge LLM used within ROOT? If so, does this calculation align with those used in the baseline methods? Is it possible that responses which appear to answer a harmful question but are actually harmless (for example, long disclaimers) are still counted as successful jailbreaks under the current metric? In addition, the paper lacks analysis of *invalid responses* generated by the Meta LLM or Judge LLM (e.g., refusal messages or nonsensical outputs). Without examining these cases, it remains unclear whether the reported ASR genuinely reflects successful jailbreaks or simply artifacts of misjudged or low-quality responses. More detailed clarification and validation are needed.

    (2) The reported ASR and bypass rate rely entirely on automated judgments, which may inherit biases from the Judge LLM and inflate the measured success rate. A fair comparison would require human evaluation or a standardized external moderation API applied uniformly to all baselines.

---

### Official Review · Reviewer_VACJ · 2025-11-01

**Soundness:** 2
**Presentation:** 2
**Contribution:** 2
**Rating:** 2
**Confidence:** 4

**Summary:**

This paper introduces ROOT, a fully automated framework for generating black-box jailbreak attacks against LLMs. The core idea is to reframe the attack as a "meta-prompt optimization" problem , where the system iteratively refines a prompt-generator (the meta-prompt) rather than a single static prompt. ROOT employs a sequential pipeline using multiple LLMs in specialized roles: a Meta LLM generates attacks , a Judge LLM evaluates responses and provides natural language "reflections" on both successes and failures , an Analysis LLM summarizes these reflections into general strategies , and an Optimize LLM integrates these "reward-guided" strategies to evolve the meta-prompt. The authors report extensive experiments showing high attack success rates (ASR), consistently above 90%, across eight major victim LLMs (including GPT-4 and Claude 3.5 Sonnet) and against standard guard models like LLaMA Guard .

**Strengths:**

The paper's primary strengths are its clarity and the significance of the problem it addresses. The problem of automated jailbreaking is significant and of clear interest to the AI safety community. The paper also demonstrates a high-quality experimental effort by testing its method across a wide breadth of eight modern LLMs . The ablation studies, such as the one validating the use of both success and failure reflections, are thorough and support the authors' design choices .

**Weaknesses:**

1. **Limited Discussion of Novelty in the Context of Prompt Optimization**: The paper's core methodology, "meta-prompt optimization", is presented as a novel approach for jailbreaking. However, this method appears to be a specific application of a well-established, general-purpose paradigm: prompt optimization. The paper does not cite or compare itself against foundational frameworks in this domain, such as DSPy[1] or TextGrad[2].

2. **Insufficient Evaluation Against State-of-the-Art Defense Methods**: The paper claims ROOT achieves "extremely high bypass success rate", but the experimental evidence for this is not sufficient. The "Guard Model Bypass Evaluation" (Table 2) only tests against LLaMA Guard and OpenAI Moderation. These are well-known but represent a standard class of static filters. The evaluation omits a more challenging and modern category of defenses. To be considered truly robust, the attack must be tested against: Self-Reminders[3], AutoDefense[4], AegisLLM[5]

[1] Khattab, Omar, et al. "Dspy: Compiling declarative language model calls into self-improving pipelines." arXiv preprint arXiv:2310.03714 (2023).

[2] Yuksekgonul, Mert, et al. "Textgrad: Automatic" differentiation" via text." arXiv preprint arXiv:2406.07496 (2024).

[3] Xie, Yueqi, et al. "Defending chatgpt against jailbreak attack via self-reminders." Nature Machine Intelligence 5.12 (2023): 1486-1496.

[4] Zeng, Yifan, et al. "Autodefense: Multi-agent llm defense against jailbreak attacks." arXiv preprint arXiv:2403.04783 (2024).

[5] Cai, Zikui, et al. "AegisLLM: Scaling Agentic Systems for Self-Reflective Defense in LLM Security." arXiv preprint arXiv:2504.20965 (2025).

**Questions:**

1.  **Regarding Novelty (Weakness 1):** The "meta-prompt optimization" loop is conceptually very similar to uncited frameworks like **DSPy** and **TextGrad**. Could you please clarify the specific novelty of the ROOT architecture compared to this existing work?

2.  **Regarding Defense Evaluation (Weakness 2):** The evaluation in Table 2 is limited to static filters. Why were more advanced, SOTA defenses like **Self-Reminders**, **AutoDefense**, or **AegisLLM** omitted?

---

### Official Review · Reviewer_Yfnj · 2025-11-01

**Soundness:** 3
**Presentation:** 3
**Contribution:** 3
**Rating:** 6
**Confidence:** 4

**Summary:**

This paper proposes ROOT (Reward-guided Meta-prompt Evolving with Reflection), a fully automated closed-loop framework for LLM jailbreak attack generation, which reframes jailbreak as a meta-prompt optimization problem. ROOT integrates Meta LLM (attack prompt generation), Judge LLM (result evaluation), Analysis LLM (strategy summarization), and Optimize LLM (meta-prompt iteration), guided by rewards from both successful and unsuccessful attack reflections. Extensive experiments across 8 mainstream LLMs (4 open-source, 6 closed-source) and 2 harmful task datasets (AdvBench, HarmBench) show ROOT achieves over 90% jailbreak success rate (ASR-GPT) universally, outperforming 12 baseline methods. It also demonstrates strong ability to bypass different Guard Models.

**Strengths:**

1. This paper proposes a training-free Jailbreak attack framework. The authors verified the effectiveness of this method through extensive experiments—its attack success rate exceeded 90% on various open-source and closed-source victim models, outperforming the existing SOTA (State-of-the-Art) level. Additionally, in the appendix experiments, the authors analyzed the iteration rounds of the meta prompt, finding that the attack success rate saturates with only 3-4 rounds.
2. The paper gives the efficiency analysis of the proposed framework. The authors conducted an analysis of token cost, and the method proposed in this paper achieves a higher attack success rate while keeping the token cost no higher than that of other baselines.
3. Modules such as Judge LLM, Analysis LLM, and Optimize LLM in this framework can be easily replaced. This means that as the capabilities of models upgrade, this framework has more potential to achieve higher performance.

**Weaknesses:**

1. There is a lack of in-depth analysis on the underlying reasons why the optimized Meta Prompt can successfully bypass the safety mechanisms of large language models (LLMs). Authors can analyze the difference in the model parameters under Meta Prompts that result in attack success or failure, including Attention mechanisms and perplexity (ppl) performance (see [1], [2], etc.). This will help analyze the operational mechanism of the Meta Prompt and provide support for future research on enhancing model security.
2. The framework lacks theoretical grounding—its optimization process is empirical and not rigorously analyzed.
3. The paper does not discuss effective defense methods against this jailbreak attack framework, which may pose potential hazards to the community. The authors need to propose effective defensive measures.

Typo:
1. line 325 “Victim LLLMs” → “Victim LLMs”
2. Figure 6 and line 464 “AST-GPT” → “ASR-GPT”


[1] Yue Liu, Xiaoxin He, Miao Xiong, Jinlan Fu, Shumin Deng, and Bryan Hooi. Flipattack: Jailbreak llms via flipping.
[2] Peng Ding, Jun Kuang, Zongyu Wang, Xuezhi Cao, Xunliang Cai, Jiajun Chen, Shujian Huang. Why Not Act on What You Know? Unleashing Safety Potential of LLMs via Self-Aware Guard Enhancement

**Questions:**

None

---

### Note · Authors · 2026-01-04

I have read and agree with the venue's withdrawal policy on behalf of myself and my co-authors.